Combined uranium-series and electron spin resonance dating from the Pliocene fossil sites of Aves and Milo’s palaeocaves, Bolt’s Farm, Cradle of Humankind, South Africa

Yu Wenjing 1 2 yuwenjing92@outlook.com
http://orcid.org/0000-0002-2905-2002 Herries Andy I. R. 1 3
Edwards Tara 4
http://orcid.org/0000-0003-1386-2243 Armstrong Brian 3 5
Joannes-Boyau Renaud 2 3
1 The Australian Archaeomagnetism Laboratory, Department of Archaeology and History, La Trobe University , Wurundjeri Country, VIC , Australia
2 Geoarchaeology and Archaeometry Research Group, Southern Cross University , Lismore, NSW , Australia
3 Palaeo–Research Institute, University of Johannesburg , Johannesburg, Gauteng , South Africa
4 Department of Geological Sciences, University of Cape Town , Rondebosch , South Africa
5 Department of Infrastructure Engineering, University of Melbourne , Melbourne , Australia
Badenhorst Shaw
Electronic publication date: 2024 Jun 28
Publication date: 2024
Volume: 12
Electronic Location ID: e17478
Received 2024 Jan 10; Accepted 2024 May 7
Copyright: © 2024 Yu et al.
Copyright year: 2024
Copyright holder: Yu et al.
License: This is an open access article distributed under the terms of the Creative Commons Attribution License, which permits unrestricted use, distribution, reproduction and adaptation in any medium and for any purpose provided that it is properly attributed. For attribution, the original author(s), title, publication source (PeerJ) and either DOI or URL of the article must be cited.
License URL: https://creativecommons.org/licenses/by/4.0/

Keywords: Combined US-ESR dating, Fossil teeth, Bolt’s Farm, Pliocene, Pleistocene, Palaeocave, Geochronology, Direct dating, South African fossils, Palaeomagnetism

Funding: Australian Research Council ARC DP140100919, ARC DP 220100195 to RJB, and ARC DP170100056 to AIRH and RJB ARC LIEF ARC LE200100022 to RJ La Trobe University Humanities and Social Science Internal Research 2021-2-HDR-0008 to WY This research was supported through the Australian Research Council discovery Grants (ARC DP140100919, ARC DP 220100195 to RJB, and ARC DP170100056 to AIRH and RJB), ARC LIEF Grants (ARC LE200100022 to RJB), a Higher Degree Research fee waiver and living scholarships from La Trobe University and the La Trobe University Humanities and Social Science Internal Research Grant 2021-2-HDR-0008 to WY. The funders had no role in study design, data collection and analysis, decision to publish, or preparation of the manuscript.

==============================
Bolt’s Farm is the name given to a series of non-hominin bearing fossil sites that have often been suggested to be some of the oldest Pliocene sites in the Cradle of Humankind, South Africa. This article reports the results of the first combined Uranium-Series and Electron Spin Resonance (US-ESR) dating of bovid teeth at Milo’s Cave and Aves Cave at Bolt’s Farm. Both tooth enamel fragments and tooth enamel powder ages were presented for comparison. US-ESR, EU and LU models are calculated. Overall, the powder ages are consistent with previous uranium-lead and palaeomagnetic age estimates for the Aves Cave deposit, which suggest an age between ~3.15 and 2.61 Ma and provide the first ages for Milo’s Cave dates to between ~3.1 and 2.7 Ma. The final ages were not overly dependent on the models used (US-ESR, LU or EU), which all overlap within error. These ages are all consistent with the biochronological age estimate (<3.4–>2.6 Ma) based on the occurrence of Stage I Metridiochoerus andrewsi. Preliminary palaeomagnetic analysis from Milo’s Cave indicates a reversal takes place at the site with predominantly intermediate directions, suggesting the deposit may date to the period between ~3.03 and 3.11 Ma within error of the ESR ages. This further suggests that there are no definitive examples of palaeocave deposits at Bolt’s Farm older than 3.2 Ma. This research indicates that US-ESR dating has the potential to date fossil sites in the Cradle of Humankind to over 3 Ma. However, bulk sample analysis for US-ESR dating is recommended for sites over 3 Ma.

Introduction

The Plio-Pleistocene locality of Bolt’s Farm, Gauteng Province, South Africa, is a 1,300 square meter area in the farthest southwestern part of the Cradle of Humankind (CoH), situated about 3 km southwest of Sterkfontein Caves and about 1 km south of Rising Star Cave (Dirks et al., 2015; Edwards et al., 2019, 2020, 2023). Bolt’s Farm is not a single site, in fact, it includes a series of active heterogeneous caves and palaeokarst deposits, many of which were mined by lime miners in the late 19th and early 20th centuries (Gommery et al., 2012; Edwards et al., 2019). Bolt’s Farm can be divided into three properties: Greensleeves in the west, Klinkert in the east, and the Main Quarry in the north (Fig. 1), however the only known fossil site in the quarry, Pit 10 (Grey Bird Pit) was destroyed by mining (Edwards et al., 2019). The significance of Bolt’s Farm is that it possesses a high density of fossil bearing deposits in a small area and these appear to represent a number of different ages, some of which have been suggested to represent some of the oldest Pliocene deposits in the CoH based on biochronology, although none have yet yielded hominin fossils (Gommery et al., 2012; Edwards et al., 2019).

Figure 1 The location and map of South Africa (A), Cradle of Humankind (B) and Bolt’s Farm (C).

The studied sites of Milo’s cave and Aves cave are indicated (Source: ESRI South Africa 50 cm colour imagery and cadastral boundaries, edited from Edwards et al., 2019). Map data: Google.

Broom (1937) first collected fossils from the area, although he referred to them as coming from Sterkfontein, in a cave around a mile south of the Australopithecus cave. The first significant phase of fossil collecting at Bolt’s Farm was undertaken by the University of California African Expedition run by C.L. Camp and F.E. Peabody in 1947–48 (Camp, 1948; Peabody, 1954). This expedition initially defined a number of fossil bearing sites as ‘pits’ (numbered up to Pit 24) (Cooke, 1991; Edwards et al., 2019). However, this early work was never fully published (Cooke, 1991; Edwards et al., 2019). Cooke (1991) published the first version of Peabody’s map of the sites from the 1940s however this was not an entirely faithful reproduction of the map which caused the later misidentification of Pit 13, Pit 15 and Pit 23 (Edwards et al., 2019). The age of these pits was historically defined based on indirect dating by biochronology based in some cases on macrofaunal and in others based on microfaunal data (Thackeray et al., 2008; Gommery et al., 2008, 2012; Edwards et al., 2019). The occurrence of Equus in some pits suggests an age of <2.33 Ma based on the currently oldest known occurrence of Equus in Africa (Geraads et al., 2004), while the occurrence of various stages in the evolution of the suid Metridiochoerus andrewsi has been used to suggest ages as young as 1.8 Ma for some deposits, and as old as the Pliocene (>2.6 Ma) for other deposits (Edwards et al., 2019, 2020; Pickford & Gommery, 2020). Moreover, bovid fossils from Pit 3 were suggested to be as young as 1.5–1.0 Ma by Reynolds (2007). However, all this material was collected from dumps back in the 1940s and so some caution should be taken in definitively associating such fauna with specific deposits, especially in areas where multiple pits are clustered close to each other; and given the results of these collections were not published at the time of collection (Cooke, 1991; Edwards et al., 2019).

Renewed work by joint French and South African teams since the 1990s has been reinvestigating these fossil deposits but has additionally identified other fossil deposits not identified or seemingly collected from in the 1940s, many also mined (Broom, 1937; Sénégas & Avery, 1998; Pickford & Gommery, 2016). Publication of this 1990 to 2000s work caused some confusion between the old pit numbers established in the 1940s, some of which also had names, and these newly defined sites (Sénégas et al., 2002; Thackeray et al., 2008; Zipfel & Berger, 2009; Gommery et al., 2012; Monson, Brasil & Hlusko, 2015; Pickford & Gommery, 2016; Edwards et al., 2019, 2020). For example, Pit 7 was called Elephant Cave (Zipfel & Berger, 2009) but was renamed Bridge Cave in recent research publications (Pickford & Gommery, 2016). Pit 16 can be seen in both the original Peabody and Cooke (1991) maps and was known as ‘Equine Pit’ but this was later given the name Milo B (aka Milo’s Pit B) as it was thought to be a newly discovered deposit (Gommery et al., 2012; Edwards et al., 2019). In contrast, Milo A (aka Milo’s pit A; Gommery et al., 2012) was not a named collecting spot from the 1940s but correlates to what Peabody called “rich bushman outcrop with bones and teeth at the surface” and Cooke (1991) referred to as “Breccia Outcrop” (Edwards et al., 2019). Due to this, in this article the name Milo’s Cave is used to refer to what has previously been called Milo A in the past. Such confusion with names and pits brings some uncertainties for biochronological dating.

While the ages for Bolt’s Farm have traditionally come from biochronology, recent Cradle wide dating studies (Pickering et al., 2019) and site specific work from Aves Cave (Edwards et al., 2020) and Waypoint 160 (Edwards et al., 2023) have shown that direct ages can be defined from these deposits through a combination of uranium-lead dating (U-Pb) and palaeomagnetism. The site of Waypoint 160, a new site identified in 2010 (Gommery et al., 2012), was originally suggested to date to 5.0–4.0 Ma based on micromammal fossils from breccia dumps outside the mined pit (Sénégas & Avery, 1998; Gommery et al., 2012; Gommery, Sénégas & Thackeray, 2014). Gommery, Sénégas & Thackeray (2014) argue that a novel species from Waypoint 160, Eurotomys bolti, is intermediate in morphology between E. pelomyoides from the 5.15 Ma site of Langebaanweg (Roberts et al., 2011) and the first true Otomyinae from the Makapansgat Limeworks between 3.0 and 2.6 Ma (Herries et al., 2013). This age estimate had significant implications for other fossils from the site such as Parapapio, which would be the oldest of this genus discovered in the region (Gommery et al., 2008, 2009). However, recent uranium-lead (U-Pb) dating and palaeomagnetism indicates the in situ deposits actually date to between 2.3 Ma and younger than 1.78 Ma, with in situ Parapapio fossils coming from towards the base of the sequence (Edwards et al., 2023). This suggests that either the micromammal correlation suggested by Sénégas & Avery (1998) is not correct, or that the E. bolti fossils that came from dump deposits are not from the Waypoint 160 deposits adjacent to them, or that a E. bolti population lasted long after other populations had evolved to become the first true Otomyinae prior to 2.6 Ma. This site highlights the issue of dating the Cradle of Humankind palaeokarst via biochronology from ex situ fossils.

ESR dating is a palaeodosimetric method that evaluates the effect of natural radioactivity recorded by a sample and has been shown to be able to date sites up to 2.6–3.0 Ma (Herries & Shaw, 2011; Herries et al., 2018). Currently, the oldest ESR dates that have been produced for the CoH come from two sites, Sterkfontein Member 4 and Drimolen Makondo. Two sets of ESR dates were produced for Sterkfontein Member 4 by Schwarcz, Grün & Tobias (1994) and then Curnoe (1999). The data produced by Schwarcz, Grün & Tobias (1994) suggested a Linear uptake (LU) model age of 2.1 ± 0.5 Ma but with bimodal peaks at 2.37 ± 0.29 and 1.72 ± 0.31 Ma and suggested that the low background dosimetry at the sites meant they were ideal for dating back into the earliest part of the Pleistocene. However, the younger peak of ages suggested that there was some mixing of different aged teeth in the sample, some from the Australopithecus africanus bearing Member 4 and others from the younger Homo and Paranthropus bearing Member 5 (Herries, Curnoe & Adams, 2009). The teeth used came from collections excavated by Alun Hughes and it is unknown whether the mixing is geologic (reworking by natural cave processes) or anthropogenic (misinterpretation of stratigraphy during excavation).

A further study by Curnoe (1999) found similar issues with Linear uptake ages ranging between 3.09 ± 0.29 and 1.23 ± 0.16 Ma for Member 4 and ages between 1.69 ± 0.26 to 0.98 ± 0.15 Ma for Member 5. Generally, the ESR ages were dismissed as being too young and unreliable. However, Herries, Curnoe & Adams (2009), Herries et al. (2013) and Herries & Shaw (2011) suggested that two issues existed in the data sets. First, a study of teeth from calcified and decalcified contexts in the Stw 53 Infill (aka Member 5A (Partridge, 2000) and named after the Stw 53 Australopithecus or early Homo cranium that was discovered in these deposits) (Curnoe & Tobias, 2006, Clarke, 2013) gave younger ages (1.35 ± 0.34 Ma) for the decalcified deposits than the calcified deposits (1.69 ± 0.20 Ma) indicating that teeth collected from decalcified contexts are likely underestimating the true age of the fossils and deposits. Second, it was clear from the location of the samples that some of the younger teeth that were thought to be from Member 4 were actually excavated from Member 5, with which their ages overlapped. The contact between Member 4 and 5 was very close at these sampling locations and it is often hard to distinguish between them. Herries & Shaw (2011) thus proposed that the ESR dates suggested a range between 2.8 and 2.1 Ma, but could be refined to between 2.58 (Gauss-Matuyama Boundary) and 2.16–2.05 Ma for Member 4 based on the reversed polarity of the flowstones within Member 4, and the occurrence of one or two short reversals in speleothem capping the deposit that was correlated to the Rèunion and/or Huckleberry Ridge events.

U-Pb dating of flowstones associated with Member 4 suggests a similar age, with flowstones capping Member 4 dating to 2.01 ± 0.06 Ma and those underlying Member 4 dating to 2.65 ± 0.30 Ma (Pickering & Kramers, 2010) but revised to 2.030 ± 0.061 and 2.645 ± 0.183 respectively by Pickering et al. (2019). Based on the relationship of the upper U-Pb age, i.e., the top of the flowstone containing the reversal events, Herries et al. (2013) and Pickering & Herries (2020) later associated the reversals to the Huckleberry Ridge event at 2.07 Ma and thus dating the Member 4 deposits and fossils to between 2.61 and 2.07 Ma (updated ages from Singer, 2014). However, some of the ESR ages suggested ages older than ~2.6 Ma and the basal speleothem age had a high uncertainty that could suggest ages as old as 2.95 Ma, more in keeping with the ESR ages (Pickering & Kramers, 2010). A unit below Member 4 that Partridge (2000) defined as Member 3 and Pickering & Kramers (2010) termed Member 2 had capping and underlying ages of 2.83 ± 0.34 (revised to 2.800 ± 0.140 Ma) and 2.80 ± 0.28 Ma (revised to 2.747 ± 0.172 Ma) respectively, also consistent with the upper range of the ESR ages. Herries (2022) thus suggested that all three methods suggest that fossil material collected from what was considered Member 4 may have ranged in age between 2.8 and 2.07 Ma and indicated that either both Member 4 and 3 had been sampled in the excavations, or that perhaps teeth had been reworked from the older units into Member 4.

Ultimately this data suggested that ESR may be reliable into the late Pliocene. Despite this the reliability of these ESR ages has been called into question based on cosmogenic ages from Sterkfontein, that suggest ages of 3.49 ± 0.19 to 3.61 ± 0.09 Ma (Granger et al., 2022). These ages are outside the range of the U-Pb that would maximally suggest an age of 2.94 Ma and the palaeomagnetism that sets an upper limit of 2.6 Ma. The oldest ESR ages from each study were 2.87 ± 0.42 and 3.09 ± 0.29 Ma suggesting upper age limits close to the lower end of the age estimates (minimally 3.3 Ma) from the cosmogenic ages (3.29 and 3.38 Ma). However, biochronology still suggests an age no older than 2.8 Ma (Frost et al., 2022), consistent with U-Pb, ESR and palaeomagnetism combined. In Member 5 at Sterkfontein the cosmogenic ages are also at odds with the ESR ages (Herries, 2022). Member 5B (aka Oldowan Infill) was dated to ~1.4 Ma with ESR, and later than a flowstone dated to 1.78 ± 0.09 Ma with U-Pb (Pickering et al., 2019). But cosmogenic nuclide burial dating has suggested an age of 2.18 ± 0.21 Ma (Granger et al., 2015). A wider study of the age of sites in the CoH indicates that there are no other sites older than 3.2 Ma (Pickering et al., 2019).

More recently, ESR has been combined with uranium-series dating to date a number of different deposits at Drimolen. Drimolen Main Quarry deposits were dated to 2.04–1.95 Ma using a combination of US-ESR, U-Pb and palaeomagnetism, again showing that these three methods correlated well in their age estimates; ~1.95 Ma based on the occurrence of the Olduvai Subchron basal reversal, 1.96 ± 0.11 Ma from U-Pb and 1.97 ± 0.15 Ma from US-ESR (Herries et al., 2020). Drimolen Makondo was dated using the same three methods and all again agreed that the deposit formed at and soon after the Gauss-Matuyama boundary at ~2.61 Ma, with basal normal polarity speleothem ages of 2.664 ± 0.392 Ma (thus >2.61 Ma), and a US-ESR age of 2.706 ± 0.428 Ma from reversed polarity deposits overlying the flowstone, thus slightly younger than 2.61 Ma (Herries et al., 2018). The Gauss-Matuyama reversal itself was identified between these two ages (Herries et al., 2018). This work confirmed that the method could be used to reliably date deposits that were at least terminal Pliocene in age, although with quite large errors. However, in this case, the final age could be refined by comparisons to other dating methods. Thus at Sterkfontein and Drimolen (Herries et al., 2018, 2020), a combination of U-Pb, ESR and palaeomagnetism has been shown to be extremely consistent in estimating the age of palaeocave deposits. In this article we build on the initial dating of fossil sites at Bolt’s Farm using U-Pb and palaeomagnetism (Pickering et al., 2019; Edwards et al., 2020, 2023) by providing the first US-ESR ages for the Aves Cave and Milo’s Cave deposits.

The studied sites

Aves cave

Aves Cave (7120889.838N, 571642.968E) is located at the northern end of the Greensleeves property, on the boundary between Greensleeves and Klinkerts and was first excavated in 2011. What was originally defined as Pits 5 (Smithy Cave), 8 (Rodent Cave), and 14 (Benchmark Pit) by Peabody was referred to as Aves Cave 4, Aves Cave 2 and Aves Cave 1 complex by Pickford & Gommery (2016), but as other names such as Smith Cave, Rodent Cave and Pit 15 in other publications (detail in: Edwards et al., 2019). Pit 15, as noted on Peabody’s map, is a small pit to the west of Pit 14 and Pit 13 appears to just be a lime miners dump. Pits 5, 8 and 14 have now been shown to have been dug into different parts of the same palaeocave deposits now simply known as ‘Aves Cave complex’ (Edwards et al., 2020).

The stratigraphy of all three pits has been described by Edwards et al. (2020), and the deposits were previously dated by U-Pb and palaeomagnetism to 3.03–2.61 Ma, making it terminal Pliocene in age. A basal flowstone in Pit 5 was U-Pb dated to 2.41 ± 0.74 Ma, while a capping flowstone in Pit 14 was U-Pb dated to 2.67 ± 0.30 Ma suggesting a depositional age not younger than 2.37 Ma and not older than 3.15 Ma. Sediments from Pit 5 and 14 have a normal magnetic polarity indicating they could not have formed younger than the Gauss-Matuyama Boundary at ~2.61–2.58 Ma (Singer, 2014; Ogg, 2020). The age is therefore sometime between ~3.03 and ~2.58 Ma in C2An.1n (Pliocene, Gauss normal Chron in geomagnetic polarity timescale), or between 3.15 and 3.12 Ma in C2An.2n, although the later age estimate was considered more unlikely (Edwards et al., 2020; Fig. 2). This age is consistent with macrofaunal remains recovered from the deposit, specifically Metridiochoerus andrewsi Stage I (Edwards et al., 2020), (also defined as Potamochoeroides hypsodon by Pickford & Gommery (2020)) which has been recovered from the Makapansgat Limeworks at 3.03–2.61 Ma and Drimolen Makondo at ~2.61 Ma, and is not known from eastern Africa before 3.4 Ma (Herries et al., 2013, 2018).

Figure 2 Aves Cave sample locations and stratigraphy.

(A) Sample locations of ESR samples (red stars with white outlines), U-Pb samples (red stars with black outlines) and paleomagnetic samples (yellow stars). (B) Sample locations and their corresponding GPTS.

Only Pit 14 was sampled for US-ESR dating and so only this stratigraphic sequence is described here (Fig. 2B). The Pit 14 sequence starts with a basal flowstone (Facies E) that is considered equivalent to the dated flowstone in Pit 5. This is overlain by Facies A that consists of large angular boulders surrounded by a fine-grained sandstone and siltstone matrix representing gravity roof collapse. This is overlain by Facies B that is a massive sandy siltstone with few clasts that represents a significant flood event. This is interlayered with thin lenses of Facies B1 that consists of sandy siltstone with more clasts and bone. Part way throughout the Pit 14 sequence Facies B1 begins to dominate and the sequence transitions to Facies F that consists of red brown laminated siltstone. The sequence is then capped by the upper 2.67 ± 0.30 Ma U-Pb dated flowstone. Bovid teeth for US-ESR dating were sampled from Facies B1. Sample ESR-01 was recovered from the base where Facies B is more prevalent just below palaeomagnetic sample AV08 (Fig. 2). ESR-02 and 03 are the ESR dated samples come from the upper part of the sequence where Facies B1 dominates between palaeomagnetism samples AV08 and AV10. The aim of this study is to conduct US-ESR dating of teeth from Aves Cave to contrast against the consistent biochronological, U-Pb and palaeomagnetic data and to test the upper age limit of US-ESR dating in the region.

Milo’s cave

Milo’s Cave (7120526.445N, 571103.527E) is situated in the middle of the Klinkert’s Property of Bolt’s Farm, just to the east of Pit 16 (Edwards et al., 2019). Because Milo’s Cave was not mined for speleothem it is a relatively intact cave remnant however it has been heavily eroded, and is covered by a Makondokarren (Fig. 3) (Brink & Partridge, 1980; Herries & Shaw, 2011). The vertical stratigraphy at the site is extremely limited, but there is significant lateral variation. The US-ESR samples were collected from a bone rich breccia deposit on the NE corner of the site that was closer to an entrance. The western part of the deposit consists of interlayered flowstone speleothem, sandstone and siltstone deposits that represent material winnowed to the rear of the cave. The ages of Milo’s cave are mostly indicated by biochronology. Gommery et al. (2012) identified Potamochoeroides hypsodon (defined by others as Metridiochoerus andrewsi Stage I) from Milo’s (A), which would suggest a late Pliocene age as with Aves Cave, however, Pickford & Gommery (2020) subsequently defined another fossil from the site as a later stage (III) of Metridiochoerus andrewsi that they use to suggest the deposit is much younger at ~1.8 Ma.

Figure 3 Field photos of Milo’s Cave.

(A) 3D photogrammetry model of Milo’s Cave showing the location of US-ESR samples 1–5. (B) Bovid tooth ESR sample embedded in situ breccia in Milo’s Cave A. (C) Site photos showing the location of US-ESR samples in the SE corner of Milo’s Cave.

Material and Methods

Sample preparation

The experiments were carried out on fossil bovid teeth extracted from breccia samples from Bolt’s Farm, South Africa. A series of tooth enamel fragments and enamel powder were prepared and analysed follow the standard ESR dating methods. The teeth used in the analysis are housed at the Ditsong National Museum of Natural History in Pretoria. Detailed information on the analysed samples of each pit is shown in Tables 1 and 2. The methodology of powder follows that established by Joannes-Boyau (2010) and fragment analysis follows Joannes-Boyau (2013), Fig. 4.

Table 1 A summary of the ages calculated by enamel fragments and enamel powder (Aves cave).

(A) Age calculated by the USESR model in the USESR program. (B) A comparison of the ages calculated by USESR, EU, and LU models. ND represents not datable, and the exact reasons for not being datable are listed in the comments. The figures in bold are approximated values because the original figures could not be modelled.

A		Ave’s cave	
Sample	AV-ESR-01	AV-ESR-02	AV-ESR-03	
Enamel											
	Dose (Gy) * (Fragment)	2,103	±	294	2,447	±	221	2,920	±	60	
	Dose (Gy) * (Powder)	1,892	±	57	1,434	±	67	1,811	±	117	
	U (parts per million (ppm))	0.23	±	0.02	0.1	±	0.1	0.11	±	0.02	
	234 U/238 U	1.2987	±	0.0460	1.1	±	0.1	1.3387	±	0.0587	
	230 Th/234 U	0.9510	±	0.0917	0.95	±	0.05	1.0009	±	0.0943	
	Thickness (mm)	1.00	±	0.02	0.90	±	0.10	1.05	±	0.05	
	Water (%)	3	±	1	3	±	1	3	±	1	
Dentine											
	U (ppm)	2.0447	±	0.0963	0.7700	±	0.2300	2.3500	±	0.1800	
	234 U/238 U	1.3810	±	0.0177	1.1	±	0.1	1.3703	±	0.0140	
	230 Th/234 U	0.8754	±	0.0272	0.95	±	0.05	0.9813	±	0.0195	
	Water (%)	5	±	3	5	±	3	5	±	3	
Sediment											
	U (ppm)	0.44	±	0.04	0.28	±	0.03	0.62	±	0.06	
	Th (ppm)	1.00	±	0.10	1.71	±	0.17	2.04	±	0.20	
	K (%)	0.04	±	0.00	0.06	±	0.00	0.13	±	0.01	
	Water (%)	15	±	10	15	±	10	15	±	10	
External dose rate sediment (μGy a −1 )										
	Beta dose	14	±	6	25	±	25	41	±	33	
	Gamma dose	532	±	41	485	±	47	485	±	47	
	Cosmic	100	±	80	100	±	80	100	±	80	
Combine US-ESR age (fragment)										
	Interal dose rate (μGy a−1)	47	±	19	22	±	22	5	±	5	
	Beta dose dentine (μGy a−1)	5	±	4	6	±	2	16	±	24	
	P enamel	0.85	±	0.32	0.94	±	0.87	3.50	±	3.74	
	P dentine	3.67	±	0.76	0.94	±	0.87	2.48	±	2.71	
	Total dose rate (μGy a−1)	698	±	65	626	±	98	631	±	195	
Age (ka) fragment	3,264	±	642	ND	ND	
Age (ka) powder:	2,915	±	402	2,322	±	357	2,759	±	498	
B	Fragments		Error	Powder		Error	Comment	
Ave’s cave	AV-ESR-01	USESR	3,264	±	642	2,915	±	402		
AV-ESR-01	LU model	3,534	±	615	3,180	±	344		
AV-ESR-01	EU model	3,152	±	530	2,837	±	280		
AV-ESR-02	USESR				2,322	±	357	Localized diagenesis, uranium leaching	
AV-ESR-02	LU model	ND	2,565	±	307		
AV-ESR-02	EU model				2,402	±	272	Localized diagenesis	
AV-ESR-03	USESR	ND	2,759	±	498		
AV-ESR-03	LU model				3,079	±	380		
AV-ESR-03	EU model				2,777	±	318		

Table 2 A summary of the ages calculated by enamel fragments and enamel powder (Milo’s Cave).

(A) Age calculated by the USESR model the USESR program. ( B) A comparison of the ages calculated by USESR, EU and LU models. ND represents not datable, and the exact reasons for not being datable are listed in the comments. NP represents not enough enamel powder volume. MA-ESR-02 is highlighted in bold to show the differences between frag and powder ages.

A		Milo’s cave	
Sample	MA-ESR-02	MA-ESR-03	MA-ESR-04	MA-ESR-05	
Enamel														
	Dose (Gy)*
(Fragments)	1,689	±	65	2,578	±	121	1,507	±	97	2,210	±	211	
	Dose (Gy)*
(Powder)	1,310	±	61		±		1,769	±	48		±		
	U (parts per million (ppm))	0.0432	±	0.0067	14.05	±	0.71	0.1530	±	0.0133	0.5525	±	0.0250	
	234 U/238 U	1.1367	±	0.1567	1.1477	±	0.043	1.2700	±	0.0583	1.3225	±	0.0825	
	230 Th/234 U	0.9451	±	0.4433	1.0276	±	0.0503	1.0119	±	0.0958	1.1827	±	0.0525	
	Thickness (mm)	0.60	±	0.05	0.85	±	0.07	0.60	±	0.05	0.61	±	0.03	
	Water (%)	3	±	1	3	±	1	3	±	1	3	±	1	
Dentine														
	U (ppm)	0.8297	±	0.0727	10.6500	±	0.3900	0.4567	±	0.0493	10.3650	±	1.0350	
	234 U/238 U	1.2197	±	0.0260	1.1207	±	0.021	1.0893	±	0.0300	1.0680	±	0.0150	
	230 Th/234 U	0.9943	±	0.0390	1.0494	±	0.0333	1.0600	±	0.0477	1.0253	±	0.0263	
	Water (%)	5	±	3	5	±	3	5	±	3	5	±	3	
Sediment														
	U (ppm)	0.94	±	0.09	1.34	±	0.13	0.36	±	0.04	0.76	±	0.08	
	Th (ppm)	2.30	±	0.23	2.8	±	0.28	1.90	±	0.19	1.47	±	0.14	
	K (%)	0.07	±	0.01	0.0784	±	0.01	0.06	±	0.01	0.07	±	0.01	
	Water (%)	15	±	10	15	±	10	15	±	10	15	±	10	
External dose rate sediment (μGy a −1 )													
	Beta dose	34	±	3	138	±	12	15	±	3	191	±	76	
	Gamma dose	532	±	41	569	±	48	486	±	46	532	±	41	
	Cosmic	100	±	80	100	±	80	100	±	80	100	±	80	
Combine US-ESR age (Fragment)													
	Interal dose rate (μGy a−1)	2	±	2	6045	±	603	89	±	11	312	±	37	
	Beta dose dentine (μGy a−1)	11	±	1	114	±	8	5	±	1	161	±	71	
	P enamel	0.66	±	0.17	−0.96	±	0.02	2.31	±	1.32	0.40	±	0.76	
	P dentine	−0.58	±	0.09	−0.97	±	0.02	1.36	±	0.96	−0.05	±	0.57	
	Total dose rate (μGy a−1)	574	±	42	6,620	±	611	638	±	43	1036	±	97	
Age (ka) fragments	2,954	±	240	ND	2,736	±	248	2,102	±	267	
Age (ka) (powder)	2,285	±	195				2,792	±	202	ND	
B		Fragments		Error	Powder		Error	Comment	
Milo’s cave	MA-ESR-02	USESR	2,954	±	240	2,285	±	195		
MA-ESR-02	LU model	3,064	±	255	2,399	±	289	
MA-ESR-02	EU model	2,951	±	239	2,310	±	273	
	MA-ESR-03	USESR	ND				High uranium concentration	
MA-ESR-03	LU model	2,599	±	455	NP	
MA-ESR-03	EU model	1,808	±	426				
MA-ESR-04	USESR	2,736	±	248	2,792	±	202	
MA-ESR-04	LU model	3,035	±	316	3,142	±	355	
MA-ESR-04	EU model	2,836	±	335	2,933	±	311	
MA-ESR-05	USESR	2,102	±	267				
MA-ESR-05	LU model	2,790	±	322	NP	
MA-ESR-05	EU model	2,086	±	246				

Figure 4 Different experimental settings for the tooth enamel fragments and powder methodology.

(A) Experimental settings for teeth that have enough powder went through both powder and fragments measurements. (B) Experimental settings for teeth went through enamel fragments measurements.

Among the seven teeth, five teeth had enough enamel powder volume. For each of these teeth, two fragments were cut from the fossil tooth, avoiding areas showing obvious diagenetic process, and cleaned from dentine and tartar using a diamond saw. One fragment was crushed into powder and split into 10 aliquots (about 50–70 mg each) then sent for gamma irradiation, while the other fragment was kept for X-ray irradiation (Fig. 4A). For the remaining two fossil teeth without enough enamel volume, only non-destructive US-ESR dating on enamel fragments was carried out (Joannes-Boyau, 2013; Yu et al., 2022). Such methods make it possible to date teeth by US-ESR dating even when the tooth samples have not met the minimum volume of tooth enamel for powder analysis (Fig. 4B).

Irradiation settings

Gamma irradiations were performed for powder samples irradiated with a calibrated 60 Co gamma source with a dose rate of 23.8 Gy/min at the Australian Nuclear Science and Technology Organisation (ANSTO), Australia. Powder samples received gamma irradiation steps of 50, 100, 250, 600, 1,200, 2,400, 4,000, 8,000, 15,000 Gy.

X-ray irradiations were performed for fragments at Southern Cross University (SCU) on a Freiberg X-ray irradiation chamber, which contains a Varian VF50. Irradiation parameters: 40KV voltage and 0.5mA current. The samples were mounted onto a Teflon sample holder, which directly exposed the fragment to the X-ray source with no shielding. The X-ray irradiation protocol followed a classic exponential dose step distribution with the dose steps: 90, 360, 900, 1,800, 3,600, 7,200 and 14,400 s (with an average dose rate of 0.22 to 0.25 Grays (Gy/s)) (details see Joannes-Boyau, 2013). The X-ray emission received by the fragment samples was calibrated using a known gamma irradiation dose performed at ANSTO.

Equivalent dose (DE) determination

ESR intensities of the aliquots were measured by a Frieberg MS5000 ESR spectrometer at SCU and all the measurements were performed with the following measurement conditions: 2 mW microwave power, 100 kHz modulation frequency with 1,024 points resolution, 0.1 mT modulation amplitude, 45 s conversion time, 12 mT sweep width, 21 s sweep time. The ESR intensities of enamel powder were determined followed Duval & Grün (2016) and the ESR intensities of enamel fragments followed Grün, Joannes-Boyau & Stringer (2008), Joannes-Boyau, Grün & Bodin (2010), Joannes-Boyau (2013). DE values and all dose response curves (DRC) were obtained by fitting a single saturated exponential (SSE) function because it provides a better fit for our data in the Matlab based fitting program McDose 2 (Joannes-Boyau, Duval & Bodin, 2018).

U-series data

High resolution U-series analyses of the enamel and dentine were performed from the remaining tooth. The concentration of 238U, 234U and 230Th was determined by ESI NW193 ArF Excimer laser ablation multi collector inductively coupled plasma mass spectrometry (LA-MC-ICP-MS). A raster or ablation track of laser ablation measurements were performed twice across both the dentine and enamel of each sample. The ablation rate of LA-ICP-MS was 20 Hz and the scan speed was 5 μm/s. 234U and 230Th were measured simultaneously, with uranium in the center faraday cup coupled with a secondary electron multiplier (SEM) and thorium on the L3 faraday cup coupled to an ion counter (IC). All other faraday cups were set to use high-gain 1,011 Ω amplifiers. The cup configuration was as follows: L3/IC (230); L2 (232); L1 (233); C/SEM (234); H1 (235); H2 (236); H3 (238). Baselines and drifts were corrected using NIST 610 and NIST 612 glass standards. In contrast, two corals (the MIS7 Faviid and MIS5 Porites corals from the Southern Cook Islands) and a rhinoceros tooth with a known concentration were used to correct 234U/238U and 230Th/238U ratios and assess the accuracy of measurements. The direction of the scan in a series of transects followed the growth axis of the enamel and dentine. An average value from the U-Th data was calculated for each dental tissue and used for the optimum U-uptake model calculation in the US-ESR model.

Age calculation

Combined US-ESR ages follow the equation: Age = DE/Da and were calculated using the USER program (Shao et al., 2012) with the following parameters: alpha-efficiency of 0.13 ± 0.02 (Grün & Katzenberger-Apel, 1994), the water content of 3 percentage by weight (wt%) in the enamel, 5 wt% with 2 wt% error in the dentine, and 0% in cementum. It is not possible to measure the water content of these open cave sites, and so a fixed value of 10% ± 5 wt% was assumed for the US-ESR age calculation. The external dose rate was determined by the radioisotopes U, Th, and K content in the attached sediments collected from the site at the Environmental Analysis Laboratory (EAL). The gamma spectrometer Inspector 1,000 was used to measure the in situ deposits surrounding the teeth at Bolt’s Farm. The cosmic dose rate was estimated by the site variation over the burial time. The error of the cosmic dose rate was set to 80% because of the complicated deposition history of Bolt’s Farm. Rn and Ra loss were not able to be measured within the dental tissue, and thus a state of equilibrium was assumed in this study. This is because the highly mineralized teeth are typically not thought to be affected by Rn and Ra loss from experience (Dirks et al., 2017; Herries et al., 2020). The mathematical algorithm for calculating the ages in the USESR program followed Shao et al. (2014). The Early Uptake (EU) model and Linear Uptake (LU) model generated by the DATA program were also applied as a reference to the age as per Grün (2009).

Results

The DE values, enamel and dentine information, external dose rate estimates for the samples, and final age estimations are presented for Aves Cave (Table 1) and Milo’s Cave respectively in Table 2. The USESR, EU, and LU models are all presented in these tables. The results provide ages ranging from 3.53 ± 0.16 to 2.32 ± 0.36 Ma depending on the tooth and the model used, and whether powder or fragment measurements were being used for Aves Cave.

Aves cave

As shown in Table 1, the US-ESR and EU models are most closely aligned with each other. In contrast LU models give slightly higher median age estimates, but the ages from all three models overlap with each other in all of the teeth. Tooth AV-ESR-02 gives the youngest ages of between ~2.6 and ~2.3 Ma (powder ages), while AV-ESR-01 and 3 give older median ages of between 3.1 and 2.8 Ma for powder and 3.5 to 3.2 Ma for fragments. However, the fragment ages have large errors and overlap within error of the powder ages.

Fragment ages were not initially able to be calculated for teeth AV-ESR-3 and AV-ESR-02. However, when this sample was tested with no isotopic error ratio allowed the generation of an unrealistic age of 4.9 million years. This is further confirmed by the AV-ESR-02 fragment age data was problematic, and cannot generate consistent results, thus exclude from the final age estimation (Table S1, Supplemental Materials). When looked at stratigraphically the sample from the deepest part of the site AV-ESR-01 give the oldest ages and the one from the youngest part gives the youngest ages AV-ESR-02. All three provide ages within the error of the U-Pb and palaeomagnetic age estimate of 3.0−2.6 Ma (Edwards et al., 2020).

Milo’s cave

As presented in Table 2, for teeth fragment ages, two of the samples (MA-ESR-02 and MA-ESR-04) give similar age estimates while the third one (MA-ESR-05) provides a much younger result. Fragment and powder ages for sample MA-ESR-04 are very similar ranging between 3.14 and 2.74 Ma with US-ESR ages being the youngest (2.79–2.74 Ma) and LU ages giving the oldest age estimates (3.14–3.04 Ma). Fragment ages for sample MA-ESR-02 are also consistent with these age estimates between 3.06 and 2.95 Ma, but powder ages are significantly younger (2.40–2.29 Ma). Age results for sample MA-ESR-03 are excluded from the final age estimations due to the high uranium concentration (See discussion).

Discussion

Before giving a final age estimation, this section will explore the reasons behind the scattered ages.

DE difference between enamel fragments and powder

Because fragment ages and powder ages are from the same tooth (same Da), the DE difference determines the age difference. This section aims to explore the DE differences between enamel fragments and powder.

The dating results indicate the systematic DE underestimation from powder to fragments. Two of three DE estimated by fragments are significantly higher (17∼30%) than the one obtained with powder (Fig. 5). All have larger discrepancies between DE than the associated errors, making each paired result statistically distinct. Except for sample MA-ESR-04, which shows similar DE for enamel powder and enamel fragments, all the other DE results obtained from enamel powder are lower than those obtained from enamel fragments.

Figure 5 Comparison between DE obtained using X-ray (fragments) and gamma irradiations (powder).

Milo’s Cave samples are indicated in red circles, and Aves Cave samples are indicated in empty blue squares.

Such differences could be due to the following reasons. Firstly, this difference could be due to the presence of the two different types of CO2-radical (AIRCORs/NOCORs). These radicals dominant the radiation-induced ESR signal in tooth enamel, and only fragment samples can measure the percentage of AICORs/NOCORs, in agreement with the previous research of ESR age systematically underestimation on enamel powder analysis (Grün, Joannes-Boyau & Stringer, 2008; Joannes-Boyau & Grün, 2011; Joannes-Boyau, 2013). In this consideration, fragments might offer a better estimate of the true age. However, at Aves Cave the enamel fragments appear to give overestimations for the age of the site when compared to U-Pb and palaeomagnetism, although they are still within error of this age estimate because of the larger uncertainties in the dates (±600 − 500 ka). Moreover, powder analysis can provide a bulk age for the enamel because the powder was divided into multiple aliquots for irradiation and measurements. In contrast, each fragment requires repeat measurement and can suffer more from potential localized diagenesis. The AV-ESR-03 fragment age likely suffers from localized diagenesis, because the calculated internal dose of AV-ESR-03 is under zero (−363 ± 810) μGy a-1 and is thus excluded from the final age estimation. Localized diagenesis is more relevant to older sites that have complicated past depositional histories, and imperfect spectra decomposition of fragment analysis can also generate large errors in US-ESR dating. Moreover, spectra decomposition, required for enamel fragments age evaluation, and imperfect spectra merging can also induce significant error. Alternatively, some fragments could be contaminated as there are huge differences between the original isotopic error ratio and no error ratio (Table S1, Supplemental Material).

Secondly, it is also worth noting that powder samples were irradiated by gamma irradiation while fragments were irradiated by X-rays. This difference in protocol explains why the DE obtained on fossil fragments is frequently higher than those obtained on powder. Gamma irradiation sources have traditionally been used as the main irradiation sources for ESR dating. Research has shown that X-ray sources are a promising irradiation source (Yu, Herries & Joannes-Boyau, 2022). However, the main problem for using an X-ray source is that calibration appears to be a much more complex task than expected. Furthermore, gamma emission is mono-energetic, while X-ray emission by most instruments would have a broader spectrum, with high to low energies photons; which complicates greatly the calibration of the source (Grün, Mahat & Joannes-Boyau, 2012). Another problem is that the ionisation difference between X-rays and Gamma rays is significant. Finally, it is also possible but unlikely that the differences, including the DE estimation, might be linked to the powdering of the sample.

Scatter ages between models

Although EU and LU models have many problems (Grün, 2009; Shao et al., 2012), ages are calculated by both the US-ESR model in the program USESR (Shao et al., 2014) and the EU model and LU model in the DATA program (Grün, 2009) for simple comparison reasons (Fig. 6). A large discrepancy between different models can be observed. The reason for the large variability in the age estimation for different samples is probably due to the complicated deposition history.

Figure 6 Age results of the USESR model compared to the Early Uptake (EU) model for the same samples.

All results show statistically indistinguishable age results between the USESR model and the EU model. Milo’s Cave is represented by red circles and Aves Cave is represented by blue circles. Ages obtained by powder were marked as hollow circles.

LU models evidently give the oldest ages and EU models seem to give a similar age estimation to the US-ESR model (Fig. 6). Yet, previous studies have preferred the LU model as their best age estimate for ESR dating in the CoH (Curnoe, 1999; Curnoe et al., 2001), correlating well with other methods at Sterkfontein (Herries, 2022). Theoretically, EU models provide what could be seen as the minimum age. It has been argued that LU models suit the CoH context better because of low background dosimetry, and the small uranium concentration in dentine (Dirks et al., 2017; Herries et al., 2020). With small uranium concentration uptake within the fossils, the difference between EU and LU is less when compared to high concentrations models. These scatter ages indicate significant uncertainties in the deposition environment (U-series ages). Bolt’s Farm has experienced important changes in terms of climatic and geological conditions. Climatic changes introduced massive erosion in the whole area and added to postdeposition in the karst events. All these processes have probably led to significant changes in the deposition environment and taphonomy process in the Bolt’s Farm area. The heterogeneity of the depositional environment possibly links to enamel modification during burial and fossilization. Some of the samples have been partly eroded out of the surrounding sediments by surface erosion and the formation of Makondo features into the deposits. While it is likely that teeth from a completely decalcified context, like a Makondo, could result in high uranium content in enamel, and thus age underestimation as seen for some samples from Sterkfontein (Herries & Shaw, 2011), it is possible that partial erosion could have similar effects for parts of the teeth. While the samples were carefully chosen in that they were still mostly encased in indurated sediments, the process of fossilisation and adjacent decalcified could have altered their geochemical parameters (e.g., density) significantly.

The fossil tooth samples from Milo’s Cave and Aves Cave have also evidently experienced some degree of uranium leaching (U-leaching), which could also be triggered by these processes. This could explain why some of the US-ESR ages from the same caves at Bolt’s Farm are producing some variation in the ages. The 230Th/234U activity ratio affects the beta dose rate calculation and U-leaching behaves as a higher 230Th/234U ratio than equilibrium in a given volume (Duval et al., 2011, 2012). Half of the analysed Bolt’s Farm dental tissues had 230Th/234U ratios higher than equilibrium, indicating that uranium had probably leached from these tissues. For this reason, sample AV-ESR-02 was excluded for age calculation due to the abnormal uranium patterns when modelling.

The complicated deposition history may also have led to the high uranium concentration in enamel. For example, MA-ESR-03 shows a significantly higher uranium concentration in enamel, at 14.05 ± 0.71 ppm. The high U-concentrations in the enamel generate the calculation of massive internal dose value, contributes to more than 89% of the total dose rate. Previous research shows low uranium concentrations in sample enamel is essential to the successful application of US-ESR dating of tooth enamel and high uranium concentrations in enamel often cause age underestimations (Duval et al., 2011, 2012). This is because high U-concentrations bring uncertainties of U-uptake and significant alpha efficiency (Grün et al., 2008) and thus it is not possible to model the correct uranium-uptake. MA-ESR-03 is thus excluded from the age results estimation for this reason.

Final age estimates

The ESR ages from Aves Cave appear to provide ages that are in stratigraphic age order in terms of median powder ages, with the lowest sample (AV-ESR-01) giving the oldest ages around 2.9 − 2.8 Ma, AV-ESR-03 the next oldest ~2.8 Ma, followed by AV-ESR-02 at 2.6 – 2.3 Ma. While there maybe issues with sample AV-ESR-02 that are causing underestimation of its age, it still falls within error of the U-Pb and palaeomagnetic chronology of Edwards et al. (2020) at 3.0 − 2.6 Ma, although only at the upper limit of their error range. Fragment ages for AV-ESR-01 suggest ages (3.5 − 3.2 Ma) older than this, although they are still within the error of the 3.0 − 2.61 Ma age estimate.

The ESR ages for Aves Cave and comparisons to other dating methods clearly indicate that ESR is able to date deposits that are at least 3.0 − 2.6 Ma in age. In terms of Milo’s Cave, which has yet to be U-Pb dated, the ages give scattered results for both powder samples ranging from ~3.1 − 2.8 and 2.4 to 2.3 Ma depending on the sample and model used to between 3.1 and 2.1 Ma for fragment ages, despite coming from the same level in the site. In this sense there is some consistency in the ages for both fragments and powder samples overall but often not on the same samples. The only sample that gives similar ages for both powder and fragment is MA-ESR-04 which suggests an age of between 3.0 and 2.7 Ma depending on the model used. The LU model gives the greatest consistency across all the fragment samples at around 3.1 to 2.8 Ma, which is again consistent with the ages for MA-ESR-04 and as such this age is likely the best age estimate.

Preliminary palaeomagnetic analysis on speleothem and sediments indicates a predominance of intermediate polarity at the site with high inclinations characteristic of reversal behaviour, but the short stratigraphic sequence at the site has made defining the nature of the polarity sequence difficult. If this intermediate polarity does suggest the occurrence of a reversal, then there are a number of reversals occurring in the time window suggested by the US-ESR at ~3.03 and 3.11 Ma. Based on the US-ESR ages the site may date to either side of the upper Kaena event reversal at ~3.11 Ma, with the flowstone underlying the fossil deposits correlating to Pickering et al’s (2019) FGI1 stage between 3.19 and 3.08 Ma and the fossil deposits dating to Pickering et al’s (2019) SED1 phase between 3.08 and 2.83 Ma. To date, the Hoogland site (Adams et al., 2010) in the very NE of the CoH and the base of AV01 have been dated to this phase of formation. If the ~3.11 Ma age is correct, this suggests that Milo’s maybe slightly older than Aves Cave, at post 3.03 Ma. However, an age of ~3.03 Ma could also be suggested.

As mentioned above, Gommery et al. (2012) initially suggested the occurrence of Metridiochoerus andrewsi Stage 1 at Milo’s Cave, which would suggest that it is of a similar age to Aves Cave and no older than 3.4 Ma based on correlations with eastern Africa. In South Africa this species is also found at Drimolen Makondo at ~2.61 Ma and the Makapansgat Limeworks at 3.03 to 2.61 Ma (Herries, 2022). The US-ESR ages are thus consistent with this biochronological age estimate. However, Pickford & Gommery (2020) have more recently suggested that fossils of a later stage (III) of Metridiochoerus andrewsi occur at Milo’s Cave, which they suggest would indicate an age of ~1.8 Ma. Pickford & Gommery (2020) note that the defining of the teeth to M. andrewsi (stage III) is based on the overall size of the canine but that it is still encased in breccia and so cannot be measured and analysed fully. Another molar was worn too much to be certain, while another molar fragment is suggested to be consistent in size with M. andrewsi stage III (Pickford & Gommery, 2020). However, a recent reanalysis of this tooth is instead consistent with M. andrewsi (Justin Adams, personal communication). The US-ESR ages are certainly more consistent with the original interpretation of Gommery et al. (2012) that Milo’s Cave dates to the late Pliocene, rather than early Pleistocene.

Conclusion

We confirm that the Bolt’s Farm cave complex represents a long-time span in the Cradle of Humankind. Despite many challenges in dating the Pliocene materials in Bolt’s Farm, powder ages appear to give the most consistent estimates (Aves Cave ~2.9 − 2.7 Ma), and the best age estimate for the deposits is considered to be between ~3.15 and 2.61 Ma deposits. In contrast, fragment ages using the LU model may provide the best age estimates for Milo’s Cave which may date to ~3.11 Ma, or sometime between ~3.1 and ~2.7 Ma. While there is some obvious complexity in both these sites it indicates that ESR can date sites that are close to, or maybe slightly older than 3 Ma, but this is best undertaken as part of a comprehensive multi-method dating strategy.

The direct dating of these two caves provides an important contribution to a better understanding of the timing of faunal occupation in Bolt’s Farm. Multiple models have been used to approach the age but the complicated stratigraphy and depositional sequences, and unknown geochemical histories (Uranium leaching) remain major challenges for producing robust radiogenic age estimates for Bolt’s Farm. US-ESR dating is a promising dating method for sites over 3 Ma, however, the fragment ages may suffer from localized diagenesis and thus bulk samples analysis is recommended to get a reliable age. Moreover, it is likely that to get a reliable age on older sites that a large number of teeth should be dated, as many may not work, and that special attention needs to be taken to assess their degree of weathering and diagenesis, even if still encased in indurated sediments. However, this could not be avoided in this case as access to samples was limited. The dating of Aves and Milo’s Caves to less than 3.2 Ma provides further evidence that there are no definitive examples of palaeocave deposits at Bolt’s Farm, or more widely in the CoH, older than 3.2 Ma as suggested by Pickering et al. (2019), at least not deposits that have been dated consistently by U-Pb, ESR and Palaeomagnetism. The suggested older ages for Australopithecus at Sterkfontein based off cosmogenic burial dating (Granger et al., 2015, 2022) remain an outlier, and one that is in contrast to the ages suggested by these combined methods at Sterkfontein itself and more broadly across the CoH (Herries, 2022).

Supplemental Information

Supplemental Information 1 Supplementary Materials.

Supplemental Information 2 A comparison of the sample ages between the original US-ESR isotopic ratio and the US-ESR isotopic ratio with a reducing factor of 100 at Bolt’s Farm.

A state of equilibrium will be reached due to the highly mineralized teeth, Rn and Ra lose are thus not taken into account in the calculation.

This work was conducted while undertaking a PhD at La Trobe University. Southern Cross University generously contributed towards the installation of the ESR spectrometer. The authors would like to thank Qingfeng Shao for his constructive advice for the USESR program. Thanks to Stephanie Potze (Formerly of the Ditsong Museum in Pretoria) for inviting us to work at the site and providing access and permits for sample collection.

Additional Information and Declarations

Competing Interests

Author Contributions

Data Availability

The authors declare that they have no competing interests.

Wenjing Yu conceived and designed the experiments, performed the experiments, analyzed the data, prepared figures and/or tables, authored or reviewed drafts of the article, and approved the final draft.

Andy I. R. Herries analyzed the data, prepared figures and/or tables, authored or reviewed drafts of the article, and approved the final draft.

Tara Edwards analyzed the data, prepared figures and/or tables, and approved the final draft.

Brian Armstrong analyzed the data, prepared figures and/or tables, and approved the final draft.

Renaud Joannes-Boyau conceived and designed the experiments, analyzed the data, authored or reviewed drafts of the article, and approved the final draft.

The following information was supplied regarding data availability:

The raw data is available at Figshare and Zenodo:

- Yu, Wenjing; Herries, Andy; Edwards, Tara; Armstrong, Brian; Joannes-Boyau, Renaud (2024). Raw data for Bolt’s Farm.zip. figshare. Dataset. https://doi.org/10.6084/m9.figshare.24945777.v1.

- Yu, W., & Armstrong, B. (2024). Sample locations and ESR raw data for Bolt’s Farm. Zenodo. https://doi.org/10.5281/zenodo.11227962.

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
