# Peer review of "Combined uranium-series and electron spin resonance dating from the Pliocene fossil sites of Aves and Milo’s palaeocaves, Bolt’s Farm, Cradle of Humankind, South Africa"

_PeerJ, doi:10.7717/peerj.17478_

## Round 0.1 · original submission · Minor Revisions

Dear authors,

I have now received the comments from the reviewers. Both suggested minor changes, and I think you will agree, the changes will only improve the paper.

I studied fauna from Bolt's Farm some years ago, and from my side as academic editor, a few comments and suggestions:

1) Thackeray et al. published a paper in 2008 on the history of research at Bolt's Farm, and I would have liked to see it cited in the Introduction, where you introduce the site and its history. The paper was published in the Annals of the Transvaal Museum. In fact, several papers appeared in this journal, and it successor, the Annals of the Ditsong National Museum of Natural History, on Bolt's Farm. I'm not necessarily suggesting to cite them all, but they do provide valuable background where you discuss the French-South African research endeavors (lines 71-78 in particular, but also elsewhere). In addition, in some cases, you seem to rely entirely on only a few key papers.

2) Lines 59-62: please specify which pits you are referring to.

3) Lines 233: it would be useful to state where the material is stored or located, for future research.

Apart from these additional comments, the paper is interesting and worth publishing.

Sincerely,
Shaw Badenhorst

Reviewer 1 ·

Basic reporting

I have read this interesting manuscript for PeerJ and think it will be a valuable contribution to the ongoing quest to date the South African paleontological sites. I have two questions that should be considered in a final version that could be suitable for publication in PeerJ.

Questions
1) How can U-series be used to date something that old? Given that 234U has a half-life of 245Ky and 230Th 75Ky, and that secular equilibrium is generally attained after six half-times given analytical uncertainties. I’m not a specialist in ESR dating, so the method should be OK, but I think that explanations must be given.
2) What is the cup configuration on the MC-ICP-MS? What are the typical voltages for 230Th and 234U using laser ablation?

Typos
Line 108 remove ± before 1.23
Line 145 double point
Line 179 what is (U-B)?
Line 344 what is AIRCORs/NOCORs?
Line 381 abbreviations for EU and LU are given too late.

Experimental design

OK

Validity of the findings

please see above

Additional comments

none

Reviewer 2 ·

Basic reporting

The article is well structured and its discussions and conclusions are in accordance with the data presented. The article has some errors that need to be corrected, but nothing too serious. The errors that need to be corrected are highlighted in the PDF that I am sending to the journal with my evaluation.

Experimental design

The article is unprecedented research that can contribute greatly to paleontological sites similar to those analyzed by the authors

Validity of the findings

The conclusions presented by the authors are in agreement with the data presented, showing the validity of the method for dating similar paleontological sites.

Annotated reviews are not available for download in order to protect the identity of reviewers who chose to remain anonymous.

---

## Round 0.2 · accepted · Accept

Dear Dr. Yu,

I am satisfied that you adequately addressed the minor comments raised by the reviewers. Your paper can therefore be accepted for publication.

However, there is one issue that you must address before publication, and it relates to the availability of the raw data. Only summaries of analyses are available and not raw data (which should be added to the supplementary materials) and models showing samples.

The latter are great but do not seem to have a digital object identifier which may make it difficult to have them available on the long-term. If Sketchfab changes URLs these will likely be difficult to track (similar issues can occur with GitHub). A DOI is made available on some repositories such as Zenodo. Both of these should be made available at the latest upon publication.

Furthermore, the dataset at https://doi.org/10.6084/m9.figshare.24945777.v1 should be mentioned in the manuscript.

sincerely,
Shaw Badenhorst